# IDENTIFYING WEIGHTS AND ARCHITECTURES OF UNKNOWN ReLU NETWORKS

## ABSTRACT

The output of a neural network depends on its parameters in a highly nonlinear way, and it is widely assumed that a network's parameters cannot be identified from its outputs. Here, we show that in many cases it is possible to reconstruct the architecture, weights, and biases of a deep ReLU network given the ability to query the network. ReLU networks are piecewise linear and the boundaries between pieces correspond to inputs for which one of the ReLUs switches between inactive and active states. Thus, first-layer ReLUs can be identified (up to sign and scaling) based on the orientation of their associated hyperplanes. Later-layer ReLU boundaries bend when they cross earlier-layer boundaries and the extent of bending reveals the weights between them. Our algorithm uses this to identify the units in the network and weights connecting them (up to isomorphism). The fact that considerable parts of deep networks can be identified from their outputs has implications for security, neuroscience, and our understanding of neural networks.

## 1 INTRODUCTION

The behavior of deep neural networks is as complex as it is powerful. The relation of individual parameters to the network's output is highly nonlinear and is generally unclear to an external observer. Consequently, it has been widely supposed in the field that it is impossible to recover the parameters of a network merely by observing its output on different inputs.

Beyond informing our understanding of deep learning, going from function to parameters could have serious implications for security and privacy. In many deployed deep learning systems, the output is freely available, but the network used to generate that output is not disclosed. The ability to uncover a confidential network not only would make it available for public use but could even expose data used to train the network if such data could be reconstructed from the network's weights.

This topic also has implications for the study of biological neural networks. Experimental neuroscientists can record some variables within the brain (e.g. the output of a complex cell in primary visual cortex) but not others (e.g. the pre-synaptic simple cells), and many biological neurons appear to be well modeled as the ReLU of a linear combination of their inputs (Chance et al., 2002). It would be highly useful if we could reverse engineer the internal components of a neural circuit based on recordings of the output and our choice of input stimuli.

In this work, we show that it is, in fact, possible in many cases to recover the structure and weights of an unknown ReLU network by querying it. Our method leverages the fact that a ReLU network is piecewise linear and transitions between linear pieces exactly when one of the ReLUs of the network transitions from its inactive to its active state. We attempt to identify the piecewise linear surfaces in input space where individual neurons transition from inactive to active. For neurons in the first layer, such boundaries are hyperplanes, for which the equations determine the weights and biases of the first layer (up to sign and scaling). For neurons in subsequent layers, the boundaries are "bent hyperplanes" that bend where they intersect boundaries associated with earlier layers. Measuring these intersections allows us to recover the weights between the corresponding neurons.

Our major contributions are:

- We identify how the architecture, weights, and biases of a network can be recovered from the arrangement of boundaries between linear regions in the network.

- We implement this procedure and demonstrate its success in recovering trained and untrained ReLU networks.
- We show that this algorithm "degrades gracefully," providing partial weights even when full weights are not recovered, and show that these situations can indicate intrinsic ambiguities in the network.

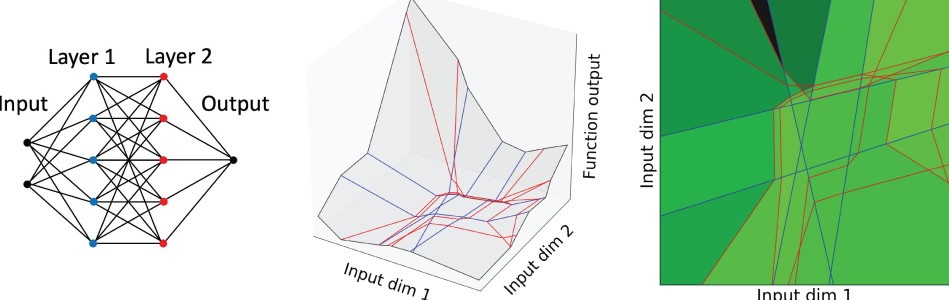

Figure 1: Left: Architecture of a ReLU network $\mathcal{N}(\mathbf{x}) : \mathbb{R}^2 \to \mathbb{R}$ with two hidden layers, each of width 5. Center: Graph of the piecewise linear function $\mathcal{N}(\mathbf{x})$ as a function of the two input variables. Right: Boundaries between linear regions of the network $\mathcal{N}$, essentially a "flattened" version of the center image. Boundaries corresponding to neurons from the first layer are shown in blue, from the second layer in red. Green shading indicates the gradient $\partial \mathcal{N} / \partial x_1$ along input dimension 1; note that the gradient is constant within each region since $\mathcal{N}(\mathbf{x})$ is piecewise linear.

## 2 RELATED WORK

Various works within the deep learning literature have considered the problem of learning a network given its output on inputs drawn (non-adaptively) from a given distribution. It is known that this problem is in general hard (Goel et al., 2017), though positive results have been found for certain specific choices of distribution in the case that the network has only one or two layers (Ge et al., 2019; Goel & Klivans, 2017). By contrast, we consider the problem of learning about a network of arbitrary depth, given the ability to issue queries at specified input points. In this work, we leverage the theory of linear regions within a ReLU network, an area that has been studied e.g. by Telgarsky (2015); Raghu et al. (2017); Hanin & Rolnick (2019a). Most recently Hanin & Rolnick (2019b) considered the boundaries between linear regions as arrangements of "bent hyperplanes". Milli et al. (2019); Jagielski et al. (2019) show the effectiveness of this strategy for networks with one hidden layer. For inference of other properties of unknown networks, see e.g. Oh et al. (2019).

Neuroscientists have long considered similar problems with biological neural networks, albeit armed with prior knowledge about network structure. For example, it is believed that complex cells in the primary visual cortex, which are often seen as translation-invariant edge detectors, obtain their invariance through what is effectively a two-layer neural network (Kording et al., 2004). A first layer is believed to extract edges, while a second layer essentially implements maxpooling. Heggelund (1981) perform physical experiments akin to our approach of identifying one ReLU at a time, by applying inputs that move individual neurons above their critical threshold one by one. Being able to solve such problems more generically would be useful for a range of neuroscience applications.

## 3 PRELIMINARIES

### 3.1 DEFINITIONS

In general, we will consider fully connected, feed-forward neural networks (multilayer perceptrons) with ReLU activations. Each such network $\mathcal{N}$ defines a function $\mathcal{N}(\mathbf{x})$ from input space $\mathbb{R}^{n_{\text{in}}}$ to output space $\mathbb{R}^{n_{\text{out}}}$. We denote the layer widths of the network by $n_{\text{in}}$ (input layer), $n_1, n_2, \ldots, n_d$, $n_{\text{out}}$ (output layer). We use $\mathbf{W}^k$ to denote the weight matrix from layer $(k-1)$ to layer $k$, where layer 0 is the input; and $\mathbf{b}^k$ denotes the bias vector for layer $k$. Given a neuron $z$ in the network, we use $z(\mathbf{x})$ to denote its preactivation for input $\mathbf{x} \in \mathbb{R}^{n_{\text{in}}}$. Thus, for the $j$th neuron in layer $k$, we have

$$z_j^k(\mathbf{x}) = \sum_{i=1}^{n_{k-1}} \mathbf{W}_{ij}^k \operatorname{ReLU}(z_i^{k-1}(\mathbf{x}) + \mathbf{b}_i^k).$$

For each neuron $z$, we will use $B_z$ to denote the set of $\mathbf{x}$ for which $z(\mathbf{x}) = 0$. In general[1], $B_z$ will be an $(n_{\text{in}} - 1)$-dimensional piecewise linear surface in $\mathbb{R}^{n_{\text{in}}}$ (see Figure 1, in which input dimension is 2 and the $B_z$ are simply lines). We call $B_z$ the *boundary* associated with neuron $z$, and we say that $B = \bigcup B_z$ is the *boundary* of the overall network. We refer to the connected components of $\mathbb{R}^{n_{\text{in}}} \setminus B$ as *regions*. Throughout this paper, we will make the *Linear Regions Assumption*: The set of regions is the set of linear pieces of the piecewise linear function $\mathcal{N}(\mathbf{x})$. While this assumption has tacitly been made in the prior literature, it is noted in Hanin & Rolnick (2019b) that there are cases where it does not hold – for example, if an entire layer of the network is zeroed out for some inputs.

## 3.2 Isomorphisms of networks

Before showing how to infer the parameters of a neural network, we must consider to what extent these parameters can be inferred unambiguously. Given a network $\mathcal{N}$, there are a number of other networks $\mathcal{N}'$ that define exactly the same function from input space to output space. We say that such networks are *isomorphic* to $\mathcal{N}$. For multilayer perceptrons with ReLU activation, we consider the following network isomorphisms:

**Permutation.** The order of neurons in each layer of a network $\mathcal{N}$ does not affect the underlying function. Formally, let $p_{k,\sigma}(\mathcal{N})$ be the network obtained from $\mathcal{N}$ by permuting layer $k$ according to $\sigma$ (along with the corresponding weight vectors and biases). Then, $p_{k,\sigma}(\mathcal{N})$ is isomorphic to $\mathcal{N}$ for every layer $k$ and permutation $\sigma$.

**Scaling.** Due to the ReLU's equivariance under multiplication, it is possible to scale the incoming weights and biases of any neuron, while inversely scaling the outgoing weights, leaving the overall function unchanged. Formally, for $z$ the $i$th neuron in layer $k$ and $c$ any positive constant, let $s_{z,c}(\mathcal{N})$ be the network obtained from $\mathcal{N}$ by replacing $\mathbf{W}^k_{\cdot i}$, $\mathbf{b}^k_i$, and $\mathbf{W}^{k+1}$ by $c\mathbf{W}^k_{\cdot i}$, $cb^k_i$, and $(1/c)\mathbf{W}^{k+1}_{i\cdot}$, respectively. It is simple to prove that $s_{z,c}(\mathcal{N})$ is isomorphic to $\mathcal{N}$ (see Appendix A).

Thus, we can hope to recover a network only up to layer-wise permutation and neuron-wise scaling. Formally, $p_{i,\sigma}(\mathcal{N})$ and $s_{z,c}(\mathcal{N})$ are generators for a group of isomorphisms of $\mathcal{N}$. (As we shall see in §5, some networks also possess additional isomorphisms.)

## 4 The algorithm

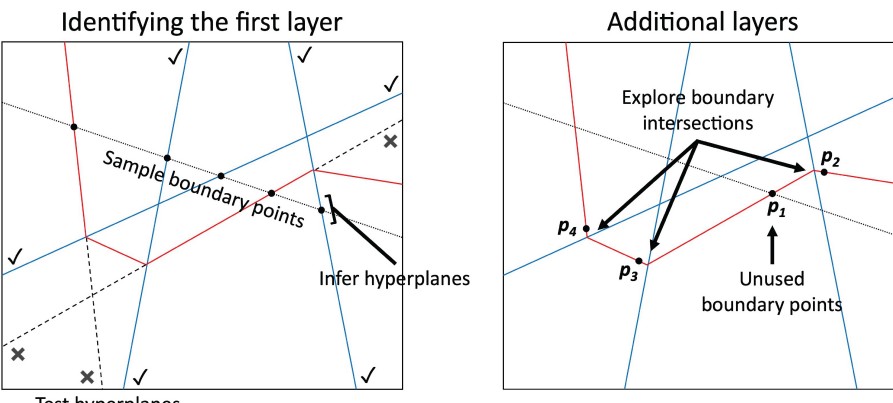

Figure 2: Schematic of our algorithms for identifying the first layer of $\mathcal{N}$ and the additional layers.

### 4.1 Intuition

Consider a network $\mathcal{N}$ and neuron $z \in \mathcal{N}$, so that $B_z$ is the boundary associated with neuron $z$. Recall that $B_z$ is piecewise linear. We say that $B_z$ *bends* at a point if $B_z$ is nonlinear at that point (that is, if the point lies on the boundary of several regions). As observed in Hanin & Rolnick (2019b), $B_z$ can bend only at points where it intersects boundaries $B_{z'}$ for $z'$ in an earlier layer of

---

[1]More precisely, this holds for all but a measure zero set of networks, and any network for which this is not true may simply be perturbed slightly.

the network. In general, the converse also holds; $B_z$ bends wherever it intersects such a boundary $B_{z'}$ (see Appendix A). Then, for any two boundaries $B_z$ and $B_{z'}$, one of the following must hold: $B_z$ bends at their intersection (in which case $z$ occurs in a deeper layer of the network), $B_{z'}$ bends (in which case $z'$ occurs in a deeper layer), or neither bends (in which case $z$ and $z'$ occur in the same layer). It is not possible for both $B_z$ and $B_{z'}$ to bend at their intersection – unless that intersection is also contained in another boundary, which is vanishingly unlikely in general. Thus, the architecture of the network can be determined by evaluating the boundaries $B_z$ and where they bend in relation to one another.

Moving beyond architecture, the weights and biases of the network can also be determined from the boundaries, one layer at a time. Boundaries for neurons in the first layer do not bend and are simply hyperplanes; the equations of these hyperplanes expose the weights from the input to the first layer (up to permutation, scaling, and sign). For each subsequent layer, the weight between neurons $z$ and $z'$ can be determined by calculating how $B_{z'}$ bends when it crosses $B_z$. The details of our algorithm below are intended to make these intuitions concrete and perform efficiently even when the input space is high-dimensional.

---

**Algorithm 1** The first layer

   Initialize $P_1 = P_2 = S_1 = \{\}$
   **for** $t = 1, \ldots, L$ **do**
      Sample line segment $\ell$
      $P_1 \leftarrow P_1 \cup \texttt{PointsOnLine}(\ell)$
   **end for**
   **for** $\mathbf{p} \in P_1$ **do**
      $H = \texttt{InferHyperplane}(\mathbf{p})$
      **if** $\texttt{TestHyperplane}(H)$ **then**
         $S_1 \leftarrow S_1 \cup \texttt{GetParams}(H)$
      **else** $P_2 \leftarrow P_2 \cup \{\mathbf{p}\}$
      **end if**
   **end for**
   **return** Parameters $S_1$,
         unused sample points $P_2$

**Algorithm 2** Additional layers

   Input $P_k$ and $S_1, \ldots, S_{k-1}$
   Initialize $S_k = \{\}$
   **for** $\mathbf{p}_1 \in P_{k-1}$ on boundary $B_z$ **do**
      Initialize $A_z = \{\mathbf{p}_1\}$, $L_z = \mathcal{H}_z = \{\}$
      **while** $L_z \not\supseteq$ Layer $k - 1$ **do**
         Pick $\mathbf{p}_i \in A$ and $\mathbf{v}$
         $\mathbf{p}', B_{z'} = \texttt{ClosestBoundary}(\mathbf{p}_i, \mathbf{v})$
         **if** $\mathbf{p}'$ on boundary **then**
            $A_z \leftarrow A_z \cup \{\mathbf{p}' + \epsilon\}$
            $L_z \leftarrow L_z \cup \{z'\}$
            $\mathcal{H}_z \leftarrow \mathcal{H}_z \cup \{\texttt{InferHyperplane}(\mathbf{p}_i)\}$
         **else** $P_k \leftarrow P_k \cup \{\mathbf{p}_1\}$; **break**
         **end if**
      **end while**
      **if** $L_z \supseteq$ Layer $k - 1$ **then** $S_k \leftarrow \texttt{GetParams}(T_z)$
      **end if**
   **end for**
   **return** Parameters $S_k$, unused sample points $P_{k+1}$

---

## 4.2 THE FIRST LAYER

We begin by identifying the first layer of the network $\mathcal{N}$, for which we must infer the number of neurons, the weight matrix $\mathbf{W}^1$, and the bias vector $\mathbf{b}^1$. As noted above, for each $z = z_i^1$ in the first layer, the boundary $B_z$ is a hyperplane with equation $\mathbf{W}_{:i}^1 \mathbf{x} + \mathbf{b}_i^1 = 0$. For each neuron $z$ in a later layer of the network, the boundary $B_z$ will, in general, bend and not be a (complete) hyperplane (see Appendix A). We may therefore find the number of neurons in layer 1 by counting the hyperplanes contained in the network's boundary $B$, and we can infer weights and biases by determining the equations of these hyperplanes.

**Boundary points along a line.** Our algorithm is based upon the identification of points on the boundary $B$. One of our core algorithmic primitives is a subroutine $\texttt{PointsOnLine}$ that takes as input a line segment $\ell \subset \mathbb{R}_{\text{in}}^n$ and approximates the set $\ell \cap B$ of boundary points along $\ell$. The algorithm proceeds by leveraging the fact that boundary points subdivide $\ell$ into regions within which $\mathcal{N}(\mathbf{x})$ is linear. We maintain a list of points in order along $\ell$ (initialized to the endpoints and midpoint of $\ell$) and iteratively perform the following operation: For each three consecutive points on our list, $\mathbf{x}_1, \mathbf{x}_2, \mathbf{x}_3$, we determine if the vectors $(\mathcal{N}(\mathbf{x}_2) - \mathcal{N}(\mathbf{x}_1))/\|\mathbf{x}_2 - \mathbf{x}_1\|_2$ and $(\mathcal{N}(\mathbf{x}_3) - \mathcal{N}(\mathbf{x}_2))/\|\mathbf{x}_3 - \mathbf{x}_2\|_2$ are equal (to within computation error) – if so, we remove the point $\mathbf{x}_2$ from our list, otherwise we add the points $(\mathbf{x}_1 + 2\mathbf{x}_2)/3$ and $(\mathbf{x}_3 + 2\mathbf{x}_2)/3$ to our list.[2] The points in the list converge by binary search to the set of discontinuities of the gradient $\partial \mathcal{N}(\mathbf{x})/\partial \mathbf{x}$, which are our desired boundary points. Note that $\texttt{PointsOnLine}$ is where we make use of our ability to query the network.

---

[2] These weighted averages speed up the search algorithm by biasing it towards points closer towards the center of the segment, which is where we expect the most intersections given our choice of segments.

**Sampling boundary points.** In order to identify the boundaries $B_z$ for $z$ in layer 1, we begin by identifying a set of boundary points with at least one on each $B_z$. A randomly chosen line segment through input space will intersect some of the $B_z$ – indeed, if it is long enough, it will intersect any fixed hyperplane with probability 1. We sample line segments $\ell$ in $\mathbb{R}_{\text{in}}^n$ and run `PointsOnLine` on each. Many sampling distributions are possible, but in our implementation we choose to sample segments of fixed (long) length, tangent at their midpoints to a sphere of fixed (large) radius. This ensures that each of our sample lines remains far from the origin, where boundaries are in closer proximity and therefore more easily confused with one another (this will become useful in the next step). Let $P_1$ be the overall set of boundary points identified on our sample line segments.

**Inferring hyperplanes.** We now proceed to fit a hyperplane to each of the boundary points we have just identified. For each $\mathbf{p} \in P_1$, there is a neuron $z$ such that $\mathbf{p}$ lies on $B_z$. The boundary $B_z$ is piecewise linear, with nonlinearities only along other boundaries, and with probability 1, $\mathbf{p}$ does not lie on a boundary besides $B_z$. Therefore, within a small enough neighborhood of $\mathbf{p}$, $B_z$ is given by a hyperplane, which we call the *local hyperplane* at $\mathbf{p}$. If $z$ is in layer 1, then $B_z$ equals the local hyperplane. The subroutine `InferHyperplane` takes as input a point $\mathbf{p}$ on a boundary $B_z$ and approximates the local hyperplane within which $\mathbf{p}$ lies. This algorithm proceeds by sampling many small line segments around $\mathbf{p}$, running `PointsOnLine` to find their points of intersection with $B_z$, and performing a linear regression to find the equation of the hyperplane containing these points.

**Testing hyperplanes.** Not all of the hyperplanes we have identified are actually boundaries for neurons in layer 1, so we need to test which hyperplanes are contained in $B$ in their entirety, and which are the local hyperplanes of boundaries that bend. The subroutine `TestHyperplane` takes as input a point $\mathbf{p}$ and a hyperplane $H$ containing that point, and determines whether the entire hyperplane $H$ is contained in the boundary $B$ of the network. This algorithm proceeds by sampling points within $H$ that lie far from $\mathbf{p}$ and applying `PointsOnLine` to a short line segment around each such point to check whether these points all lie on $B$. Applying `TestHyperplane` to those hyperplanes inferred in the preceding step allows us to determine those $B_z$ for which $z$ is in layer 1.

**From hyperplanes to parameters.** Finally, we identify the first layer of $\mathcal{N}$ from the equations of hyperplanes contained in $B$. The number of neurons in layer 1 is given simply by the number of distinct $B_z$ which are hyperplanes. As we have observed, for $z = z_i^1$ in layer 1, the hyperplane $B_z$ is given by $\mathbf{W}_{\cdot i}^1 \mathbf{x} + \mathbf{b}_i^1 = 0$. We can thus determine $\mathbf{W}_{\cdot i}^1$ and $\mathbf{b}_i^1$ up to multiplication by a constant. However, we have already observed that scaling $\mathbf{W}_{\cdot i}^1$ and $\mathbf{b}_i^1$ by a positive constant (while inversely scaling $\mathbf{W}_{i \cdot}^2$) is a network isomorphism (§3.2). Therefore, we need only determine the true sign of the multiplicative constant, corresponding to determining which side of the hyperplane is zeroed out by the ReLU. This determination of sign will be performed below in §4.3.

**Sample complexity.** We expect the number of queries necessary to obtain weights and biases (up to sign) for the first layer should grow as $O(n_{\text{in}}(\sum_i n_i) \log n_1)$, which for constant-width networks is only slightly above the number of parameters being inferred. Assuming that biases in the network are bounded above, each sufficiently long line has at least a constant probability of hitting a given hyperplane, suggesting that $\log n_1$ lines are required according to a coupon collector-style argument. Hanin & Rolnick (2019a) show that under natural assumptions, the number of boundary points intersecting a given line through input space grows linearly in the total number of neurons in the network. Finally, each boundary point on a line requires $O(n_{\text{in}})$ queries in order to fit a hyperplane.

### 4.3 Additional layers

We now assume that the weights $\mathbf{W}^1, \ldots, \mathbf{W}^{k-1}$ and biases $\mathbf{b}^1, \ldots, \mathbf{b}^{k-1}$ have already been determined within the network $\mathcal{N}$, with the exception of the sign choice for weights and biases at each neuron in layer $k-1$. We now show how it is possible to determine the weights $\mathbf{W}^k$ and biases $\mathbf{b}^k$, along with the correct signs for $\mathbf{W}^{k-1}$ and $\mathbf{b}^{k-1}$.

**Closest boundary along a line.** In this part of our algorithm, we will need the ability to move along a boundary to its intersection with another boundary. For this purpose, the subroutine `ClosestBoundary` will be useful. It takes as input a point $\mathbf{p}$, a vector $\mathbf{v}$ and the network parameters as determined up to layer $k-1$, and outputs the smallest $c > 0$ such that $\mathbf{q} = \mathbf{p} + c\mathbf{v}$ lies on $B_z$ for some $z$ in layer at most $k-1$. In order to compute $c$, we consider the region $R$ within which $\mathbf{p}$ lies, which is associated with a certain pattern of active and inactive ReLUs. For each boundary $B_z$, we can calculate the hyperplane equation which would define $B_z$ were it to intersect $R$, due to the fixed pattern of active and inactive neurons within $R$, and we can calculate the distance

from $\mathbf{p}$ to this hyperplane. While not every boundary $B_z$ intersects $R$, the closest boundary does, allowing us to find the desired $c$.

**Unused boundary points.** In order to identify the boundaries $B_z$ for $z$ in layer $k$, we wish to identify a set of boundary points with at least one on each such boundary. However, in previous steps of our algorithm, a set $P_{k-1}$ of boundary points was created, of which some were used in ascertaining the parameters of earlier layers. We now consider the subset $P_k \subset P_{k-1}$ of points that were not found to belong to $B_z$, for $z$ in layers 1 through $k-1$. These points have already had their local hyperplanes determined.

**Exploring boundary intersections.** Consider a point $\mathbf{p}_1 \in P_k$ such that $\mathbf{p}_1 \in B_z$. Note that $B_z$ will, in general, have nonlinearities where it intersects each $B_{z'}$ for which $z'$ lies in an earlier layer than $z$. We explore these intersections, and in particular attempt to find a point of $B_z \cap B_{z'}$ for every $z'$ in layer $k-1$. Given the local hyperplane $H$ at $\mathbf{p}_1$, we pick a direction $\mathbf{v}$ along $H$ and apply ClosestBoundary to calculate the closest point of intersection $\mathbf{p}'$ with $B_{z'}$ for all $z'$ already identified in the network. (Below we discuss how best to pick $\mathbf{v}$.) Note that if $z$ is in layer $k$, then $\mathbf{p}'$ must be on $B_z$ as well as $B_{z'}$, while if $z$ is in a later layer of the network, then there may exist unidentified neurons in layers below $z$ and therefore $B_z$ may bend before meeting $B_{z'}$. We check if $\mathbf{p}'$ lies on $B_z$ by applying PointsOnLine, and if so apply InferHyperplane to calculate the local hyperplane of $B_z$ on the other side of $B_{z'}$ from $\mathbf{p}_1$. We select a representative point $\mathbf{p}_2$ on this local hyperplane. We repeat the process of exploration from the points $\mathbf{p}_1, \mathbf{p}_2, \ldots$ until one of the following occurs: (i) a point of $B_z \cap B_{z'}$ has been identified for every $z'$ in layer $k-1$ (this may be impossible; see §5), (ii) $z$ is determined to be in a layer deeper than $k$ (as a result of $\mathbf{p}'$ not lying on $B_z$), or (iii) a maximum number of iterations has been reached.

**How to explore.** An important step in our algorithm is exploring points of $B_z$ that lie on other boundaries. Given a set of points $A_z = \{\mathbf{p}_1, \mathbf{p}_2, \ldots, \mathbf{p}_m\}$ on $B_z$, we briefly consider several methods for picking a point $\mathbf{p}_i$ and direction $\mathbf{v}$ along the local hyperplane at $\mathbf{p}_i$ to apply ClosestBoundary. One approach is to pick a random point $\mathbf{p}_i$ from those already identified and a random direction $\mathbf{v}$; this has the advantage of simplicity. However, it is somewhat faster to consider for which $z'$ the intersection $B_z \cap B_{z'}$ has not yet been identified and attempt specifically to find points on these intersections. One approach for this is to pick a missing $z'$ and identify for which $\mathbf{p}_i$ the boundary $B_{z'}$ lies on the boundary of the region containing $\mathbf{p}_i$ and solve a linear program to find $\mathbf{v}$. Another approach is to pick a missing $z'$ and a point $\mathbf{p}_i$, calculate the hyperplane $H$ which would describe $B_{z'}$ under the activation pattern of $\mathbf{p}_i$, and choose $\mathbf{v}$ along the local hyperplane to $\mathbf{p}_i$ such that the distance to $H$ is minimized. This is the approach which we take in our implementation, though more sophisticated approaches may exist and present an interesting avenue for further work.

**From boundaries to parameters.** We now identify layer $k$ of $\mathcal{N}$, along with the sign of the parameters of layer $k-1$, by measuring the extent to which hyperplanes bend at their intersection. We are, in addition, able to identify the correct signs at layer $k-1$ by solving an overconstrained system of constraints capturing the influence of neurons in layer $k-1$ on different regions of input space. The following theorem formalizes the inductive step that allows us to go from what we know at layer $k-1$ (weights and biases, up to scaling and sign) to the equivalent set of information for layer $k$, plus filling in the signs for layer $k-1$. The proof is given in Appendix B.

**Theorem.** *The following holds true for deep multi-layer perceptrons $\mathcal{N}$ satisfying the Linear Region Assumption (§3.1), excluding a set of networks with measure zero:*

*Suppose that the weights and biases of $\mathcal{N}$ are known up through layer $k-1$, with the exception that for each neuron in layer $k-1$, the sign of the incoming weights and the bias is unknown. Suppose also that for each $z$ in layer $k$, there exists an ordered set of points $A_z = \{\mathbf{p}_1, \mathbf{p}_2, \ldots, \mathbf{p}_m\}$ such that: (i) Each point lies on the boundary of $B_z$, and in (the interior of) a distinct region with respect to the earlier-layer boundaries already known; (ii) each point (except for $\mathbf{p}_1$) has a precursor in an adjacent region; (iii) for each such pair of points, the local hyperplanes of $B_z$ are known, as is the boundary $B_{z'}$ dividing them ($z'$ in an earlier layer); (iv) the set of such $z'$ includes all of layer $k-1$.*

*Then, it is possible to recover the weights and biases for layer $k$, with the exception that for each neuron, the sign of the incoming weights and the bias is unknown. It is also possible to recover the sign for every neuron in layer $k-1$.*

Note that even when assumption (iv) of the Theorem is violated, the algorithm recovers the weights corresponding to whichever boundaries are successfully crossed (as we verify empirically in §6).

## 5 DISCUSSION

We here explore some reasons why our algorithm may fail, motivate our recursive approach, and discuss the potential for generalizations to different architectures.

**Non-intersecting boundaries.** It is possible that for some neurons $z$ and $z'$ in consecutive layers, there is no point of intersection between the boundaries $B_z$ and $B_{z'}$ (or that this intersection is very small), making it impossible to infer the weight between $z$ and $z'$ by our algorithm. Some such cases represent an ambiguity in the underlying network – an additional isomorphism to those described in §3.2. Namely, $B_z \cap B_{z'}$ is empty if one of the following cases holds: (1) whenever $z$ is active, $z'$ is inactive; (2) whenever $z$ is active, $z'$ is active; (3) whenever $z$ is inactive, $z'$ is inactive; or (4) whenever $z$ is inactive, $z'$ is active. In case 1, observe that a slight perturbation to the weight $w$ between $z$ and $z'$ has no effect upon the network's output; thus $w$ is not uniquely determined. Cases 2-4 present a more complicated picture; depending on the network, there may or may not be additional isomorphisms.

**Boundary topology.** For simplicity in our algorithm, we have not considered the relatively rare cases where boundaries $B_z$ are disconnected or bounded. If $B_z$ is disconnected, then it may not be possible to find a connected path along it that intersects all boundaries arising from the preceding layer. In this case, it is simple to infer that two independently identified pieces of the boundary belong to the same neuron to infer the full weight vector. Next, if $B_z$ is bounded for some $z$, then it is a closed $(d-1)$-dimensional surface within $d$-dimensional input space[3]. While our algorithm requires no modification in this case, bounded $B_z$ may be more difficult to find by intersection with randomly chosen lines, and a more principled sampling method may be helpful.

**Our recursive approach.** Our approach proceeds layer by layer, leveraging the fact that each boundary bends only those for those boundaries corresponding to earlier neurons in the network. Our approach in the first layer is, however, distinct from (and simpler than) the algorithm for subsequent layers. One might wonder why, once the first $k-1$ layers have been identified, it is not possibly simply to apply our first-layer algorithm to the $n_{k-1}$-dimensional "input space" arising from activations of layer $k-1$. Unfortunately, this is not possible in general, as this would require the ability to evaluate layer $k$ for arbitrary settings of layer $k-1$. ReLU networks are hard to invert, and therefore it is unclear how one could manufacture an input for a specified layer $k-1$ activation, even while knowing the parameters for the first $k-1$ layers.

**Other architectures.** While we have expressed our algorithm in terms of multilayer perceptrons with ReLU activation, it also extends to various other architectures of neural network. Other piecewise linear activation functions admit similar algorithms. For a network with convolutional layers, it is possible to use the same approach to infer the weights between neurons, with two caveats: (i) As we have stated it, the algorithm does not account for weight-sharing – the number of "neurons" in each layer is thus dependent on the input size, and is very large for reasonably sized images. (ii) Pooling layers *do* affect the partition into activation regions, and indeed introduce new discontinuities into the gradient; our algorithm therefore does not apply. For skip connections as in ResNets (He et al., 2016), our algorithm holds with slight modification, which we defer until future work.

## 6 EXPERIMENTS

We verified the success of our algorithm on both untrained and trained networks. In keeping with literature on ReLU network initialization (He et al., 2015; Hanin & Rolnick, 2018), networks were initialized using i.i.d. normal weights with variance 2/fan-in and i.i.d. normal biases with unit variance. We trained networks on either the MNIST dataset or a memorization task of 1000 "datapoints" of dimension 10 with coordinates drawn i.i.d. from a unit Gaussian and given arbitrary binary labels. Training was performed using the Adam optimizer and a cross-entropy loss applied to the softmax of the final layer, over 20 epochs for MNIST and 1000 epochs for the memorization task. The trained networks (when sufficiently large) were able to attain near-perfect accuracy. We observed that both the first-layer algorithm and additional-layer algorithm identified weights and biases to within extremely high accuracy (see Figures 3 and 4). Even in cases where, for the additional-layer algorithm, a small fraction of neurons were not identified (see §5), the algorithm was able to correctly predict the remaining parameters.

---

[3]For 2D input, such $B_z$ must be topological circles, but for higher dimensions, it is conceivable for them to be more complicated surfaces, such as toroidal polyhedra.

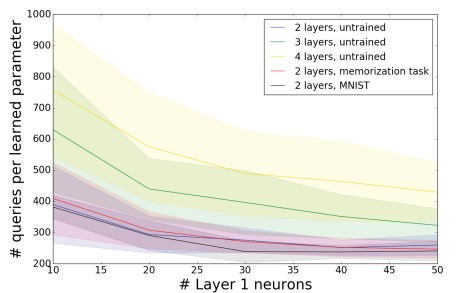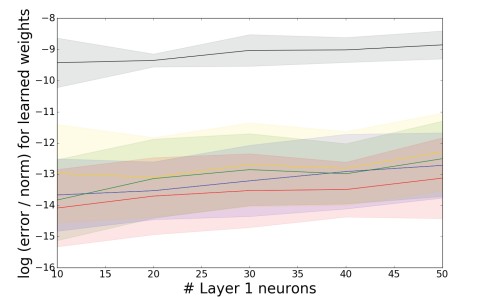

Figure 3: Results of our first-layer algorithm, applied to networks with two or more hidden layers as the width of the first layer varies. All other layers have fixed width 10. Untrained networks have input and output dimension 10, those trained on the memorization task have input dimension 10 and output dimension 2, and those trained on MNIST have input dimension 784 and output dimension 10. Left: The number of queries issued by our algorithm per parameter identified in the network $\mathcal{N}$; the algorithm is terminated once the desired number of neurons have been identified. Right: Log normalized error $\log(||\hat{\mathbf{W}}^1 - \mathbf{W}^1||_2/||\hat{\mathbf{W}}^1||_2)$ for $\hat{\mathbf{W}}^1$ the approximated weights. Weight vectors were scaled to unit norm to account for isomorphism (see §3.2). Curves are averaged over 5 runs in the case of MNIST and 40 runs otherwise, with standard deviations shown.

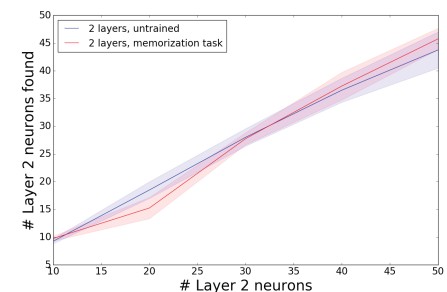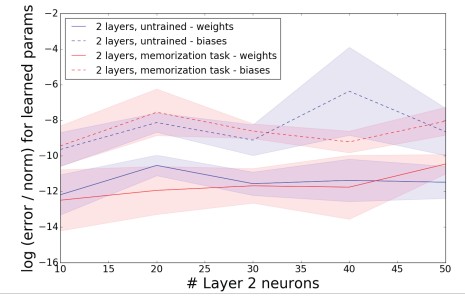

Figure 4: Results of our algorithm for additional layers, applied to networks with two layers, the first layer of width 10, as the width of the second layer varies. Left: Number of estimated layer 2 neurons. Right: Log normalized error between estimated and corresponding true neurons (as in Figure 3 above) for approximated weights $\hat{\mathbf{W}}^1$ and biases $\hat{\mathbf{b}}^1$. Curves are averaged over 4 runs, with standard deviations between runs shown as shaded regions.

## 7 CONCLUSION

In this work, we have shown that it is often possible to recover the architecture, weights, and biases of deep ReLU networks by repeated queries. We proceed by identifying the boundaries between linear regions of the network and the intersections of these boundaries. Our approach is theoretically justified and empirically validated on networks before and after training. Where the algorithm does not succeed in giving a complete set of weights, it is nonetheless able to give a partial set of weights, and incompleteness in some cases reflects unresolvable ambiguities about the network.

Our approach works for a wide variety of networks, though not all. It is limited to ReLU or otherwise piecewise linear activation functions, though we believe it possible that a continuous version of this method could potentially be developed in future work for use with sigmoidal activation. If used with convolutional layers, our method does not account for the symmetries of the network and therefore scales with the size of the input as well as the number of features, resulting in high computation. Finally, the method is not robust to defenses such as adding noise to the outputs of the network, and therefore can be thwarted by a network designer that seeks to hide their weights/architecture.

We believe that the methods we have introduced here will lead to considerable advances in identifying neural networks from their outputs, both in the context of deep learning and, more speculatively, in neuroscience. While the implementation we have demonstrated here is effective in small instances, we anticipate future work that optimizes these methods for efficient use with different architectures and at scale.

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

## A  USEFUL LEMMATA

**Lemma 1** (Isomorphism under scaling). *Given an MLP $\mathcal{N}$ with ReLU activation, the network $s_{z,c}(\mathcal{N})$ is isomorphic to $\mathcal{N}$ for every neuron $z$ and constant $c > 0$.*

*Proof.* Suppose that $z = z_i^k$ is the $i$th neuron in layer $k$. Then, for each neuron $z_j^{k+1}$ in layer $k+1$ of the network $\mathcal{N}$, we have:

$$
\begin{aligned}
z_j^{k+1}(\mathbf{x}) &= \sum_{i=1}^{n_k} \mathbf{W}_{ij}^k \operatorname{ReLU}(z_i^k(\mathbf{x}) + \mathbf{b}_i^k) \\
&= \sum_{i=1}^{n_k} \mathbf{W}_{ij}^k \operatorname{ReLU}\left( \left( \sum_{h=1}^{n_{k-1}} \mathbf{W}_{hi}^{k-1} \operatorname{ReLU}(z_h^{k-1}(\mathbf{x}) + \mathbf{b}_h^{k-1}) \right) + \mathbf{b}_i^k \right)
\end{aligned}
\tag{1}
$$

By comparison, in network $s_{z,c}(\mathcal{N})$, we have:

$$
\begin{aligned}
z_j^{k+1}(\mathbf{x}) &= \sum_{i=1}^{n_k} \frac{1}{c} \mathbf{W}_{ij}^k \operatorname{ReLU}(z_i^k(\mathbf{x}) + c\mathbf{b}_i^k) \\
&= \sum_{i=1}^{n_k} \frac{1}{c} \mathbf{W}_{ij}^k \operatorname{ReLU}\left( \left( \sum_{h=1}^{n_{k-1}} c\mathbf{W}_{hi}^{k-1} \operatorname{ReLU}(z_h^{k-1}(\mathbf{x}) + \mathbf{b}_h^{k-1}) \right) + c\mathbf{b}_i^k \right) \\
&= \sum_{i=1}^{n_k} \mathbf{W}_{ij}^k \operatorname{ReLU}\left( \left( \sum_{h=1}^{n_{k-1}} \mathbf{W}_{hi}^{k-1} \operatorname{ReLU}(z_h^{k-1}(\mathbf{x}) + \mathbf{b}_h^{k-1}) \right) + \mathbf{b}_i^k \right),
\end{aligned}
\tag{2}
$$

where we used the property that $\operatorname{ReLU}(cx) = c \operatorname{ReLU}(x)$ for any $c > 0$.

As expressions (1) and (2) are equal, we conclude that $s_{z,c}(\mathcal{N})$ is isomorphic to $\mathcal{N}$. $\qquad\square$

**Lemma 2** (Bending hyperplanes). *The set of networks $\mathcal{N}$ with the following property has measure zero in the space of networks: There exist neurons $z_i^{k-1}$ and $z_j^k$ in consecutive layers such that the boundary $B_{z_j^k}$ intersects $B_{z_i^{k-1}}$ but does not bend at the intersection.*

*Proof.* Observe that $B_{z_j^k}$ is defined by the equation:

$$
0 = z_j^k(\mathbf{x}) = \sum_{h=1}^{n_{k-1}} \mathbf{W}_{hj}^k \operatorname{ReLU}(z_h^{k-1}(\mathbf{x}) + \mathbf{b}_h^{k-1}).
$$

As it does not bend when it intersects $B_{z_i^{k-1}}$, the gradient of the RHS must remain unchanged when $\operatorname{ReLU}(z_i^{k-1}(\mathbf{x}) + \mathbf{b}_i^{k-1})$ flips between active and inactive. Unless another neuron transitions simultaneously with $z_i^{k-1}$ (an event that occurs with measure zero), this can happen only if $\mathbf{W}_{ij}^k = 0$, which itself is a measure zero event. $\qquad\square$

## B  PROOF OF THEOREM

In this proof, we will show how the information we are given by the assumptions of the theorem is enough to recover the weights and biases for each neuron $z$ in layer $k$. We will proceed for each $z$ individually, progressively learning weights between $z$ and each of the neurons in the preceding layer (though for skip connections this procedure could also easily be generalized to learn weights from $z$ to earlier layers).

For each of the points $\mathbf{p}_i \in A_z$, suppose that $H_i$ is the local hyperplane associated with $\mathbf{p}_i$ on boundary $B_z$. The gradient $\frac{\partial z}{\partial \mathbf{x}}(\mathbf{p}_i)$ at $\mathbf{p}_i$ is orthogonal to $H_i$, and we thus already know the direction of the gradient, but its magnitude is unknown to us. We will proceed in order through the points $\mathbf{p}_1, \mathbf{p}_2, \ldots, \mathbf{p}_m$, with the goal of identifying $\frac{\partial z}{\partial \mathbf{x}}(\mathbf{p}_i)$ for each $\mathbf{p}_i$, up to a single scaling factor, as this computation will end up giving us the incoming weights for $z$.

We begin with $\mathbf{p}_1$ by assigning $\frac{\partial z}{\partial \mathbf{x}}(\mathbf{p}_1)$ arbitrarily to either one of the two unit vectors orthogonal to $H_i$. Due to scaling invariance (Lemma 1), the weights of $\mathcal{N}$ can be rescaled without changing the function so that $\frac{\partial z}{\partial \mathbf{x}}(\mathbf{p}_i)$ is multiplied by any positive constant. Therefore, our arbitrary choice can be wrong at most in its sign, and we need not determine the sign at this stage. Now, suppose towards induction that we have identified $\frac{\partial z}{\partial \mathbf{x}}(\mathbf{p}_i)$ (up to sign) for $i = 1, \ldots, s-1$. We wish to identify $\frac{\partial z}{\partial \mathbf{x}}(\mathbf{p}_s)$.

By assumption (ii), there exists a precursor $\mathbf{p}_r$ to $\mathbf{p}_s$ such that $H_r$ and $H_s$ intersect on a boundary $B_{z'}$. Let $\mathbf{v}_r = t_z \frac{\partial z}{\partial \mathbf{x}}(\mathbf{p}_r)$ be our estimate of $\frac{\partial z}{\partial \mathbf{x}}(\mathbf{p}_r)$, for unknown sign $t_z \in \{+1, -1\}$. Let $\mathbf{v}_s$ be a unit normal vector to $H_s$, so that $\mathbf{v}_s = ct_z \frac{\partial z}{\partial \mathbf{x}}(\mathbf{p}_s)$ for some unknown constant $c$. We pick the sign of $\mathbf{v}_s$ so that it has the same orientation as $\mathbf{v}_r$ with respect to the surface $B_z$, and thus $c > 0$. Finally, let $\mathbf{v} = t_{z'} \frac{\partial z'}{\partial \mathbf{x}}(\mathbf{p}_r) = t_{z'} \frac{\partial z'}{\partial \mathbf{x}}(\mathbf{p}_s)$ be our estimate of the gradient of $z'$; where $t_{z'} \in \{+1, -1\}$ is also an unknown sign (recall that since $z'$ is in layer $k - 1$ we know its gradient up to sign). We will use $\mathbf{v}$ and $\mathbf{v}_r$ to identify $\mathbf{v}_s$.

Suppose that $z = z_j^k$ is the $j$th neuron in layer $k$ and that $z' = z_h^{k-1}$ is the $h$th neuron in layer $k - 1$. Recall that

$$z(\mathbf{x}) = z_j^k(\mathbf{x}) = \sum_{i=1}^{n_{k-1}} \mathbf{W}_{ij}^k \operatorname{ReLU}(z_i^{k-1}(\mathbf{x}) + \mathbf{b}_i^{k-1}). \tag{3}$$

As $B_{z'}$ is the boundary between inputs for which $z' = z_h^{k-1}$ is active and inactive, $\operatorname{ReLU}(z_h^{k-1}(\mathbf{x}) + \mathbf{b}_h^{k-1})$ must equal zero either (Case 1) on $H_r$ or (Case 2) on $H_s$.

In Case 1, we have

$$\frac{\partial z}{\partial \mathbf{x}}(\mathbf{p}_s) - \frac{\partial z}{\partial \mathbf{x}}(\mathbf{p}_r) = \mathbf{W}_{hj}^k \frac{\partial z'}{\partial \mathbf{x}}(\mathbf{p}_r),$$

or equivalently:

$$ct_z \mathbf{v}_s - t_z \mathbf{v}_r = \mathbf{W}_{hj}^k t_{z'} \mathbf{v},$$

which gives us the equation:

$$c\mathbf{v}_s - \mathbf{v}_r = \mathbf{W}_{hj}^k t_z t_{z'} \mathbf{v}.$$

Since we know the vectors $\mathbf{v}_s, \mathbf{v}_r, \mathbf{v}$, we are able to deduce the constant $c$.

A similar equation arises in Case 2:

$$\mathbf{v}_r - c\mathbf{v}_s = \mathbf{W}_{hj}^k t_z t_{z'} \mathbf{v},$$

giving rise to the same value of $c$. We thus may complete our induction. In the process, observe that we have calculated a constant $\mathbf{W}_{hj}^k t_z t_{z'} t'$, where the sign $t'$ is $+1$ in Case 1 and $-1$ in Case 2. Note that $t_{z'} t'$ can be calculated based on whether $\mathbf{v}$ points towards $\mathbf{p}_r$ or $\mathbf{p}_s$. Therefore, we have obtained $\mathbf{W}_{hj}^k t_z$, which is exactly the weight (up to $z$-dependent sign) that we wished to find. Once we have all weights incoming to $z$ (up to sign), it is simple to identify the bias for this neuron (up to sign) by calculating the equation of any known local hyperplane for $B_z$ and using the known weights and biases from earlier layers.

To complete the proof, we must now also calculate the correct signs $t_{z'}$ of the neurons in layer $k-1$. Pick some $z = z_j^k$ in layer $k$ and observe that for all points $\mathbf{p}_s \in A_z$ there corresponds an equation, obtained by taking gradients in equation (3):

$$\frac{\partial z_j^k}{\partial \mathbf{x}}(\mathbf{p}_s) = \sum_{i=1}^{n_{k-1}} \mathbf{W}_{ij}^k \mathbb{1}_{i,s} \frac{\partial z_i^{k-1}}{\partial \mathbf{x}}(\mathbf{p}_s),$$

where $\mathbb{1}_{i,s}$ equals 1 if $\mathbf{p}_s$ is on the active side of $B_{z_i^{k-1}}$. We can substitute in our (sign-unknown) values for these various quantities:

$$t_z \mathbf{v}_s = \sum_{i=1}^{n_{k-1}} \mathbf{W}_{ij}^k \mathbb{1}_{i,s} t_{z_i^{k-1}} \mathbf{v}_i.$$

Now, we may estimate $\mathbb{1}_{i,s}$ by a function $\mathbb{1}'_{i,s}$ that is 1 if $\mathbf{p}_s$ and $\mathbf{v}_i$ are on the same side of $B_{z_i^{k-1}}$. This estimate will be wrong exactly when $t_{z_i^{k-1}} = -1$. Thus, $\mathbb{1}_{i,s} = (1 + t_{z_i^{k-1}} \mathbb{1}'_{i,s})/2$, giving us the equation:

$$t_z \mathbf{v}_s = \sum_{i=1}^{n_{k-1}} \mathbf{W}_{ij}^k \frac{1 + t_{z_i^{k-1}} \mathbb{1}'_{i,s}}{2} t_{z_i^{k-1}} \mathbf{v}_i$$

$$= \frac{1}{2} \sum_{i=1}^{n_{k-1}} \mathbf{W}_{ij}^k (t_{z_i^{k-1}} + \mathbb{1}'_{i,s}) \mathbf{v}_i$$

All the terms of this equation are known, with the exception of $t_z$ and the $n_{k-1}$ variables $t_{z_i^{k-1}}$ – giving us a linear system in $n_{k-1} + 1$ variables. For a given $z_j^k$, there are $n_{k-1}$ different $\mathbf{p}_s$ representing the intersections with $B_{z'}$ for each $z'$ in layer $k - 1$; choosing these $\mathbf{p}_s$ should in general give linearly independent constraints. Moreover, the equation is in fact a vector equality with dimension $n_{\text{in}}$; hence, it is a highly overconstrained system, enabling us to identify the signs $t_{z_i^{k-1}}$ for each $z_i^{k-1}$. This completes the proof of the theorem.

