# OpenReview forum: "Identifying Weights and Architectures of Unknown ReLU Networks"
_ICLR.cc/2020/Conference — Reject_

### Official Review · AnonReviewer1 · 2019-10-22
**Official Blind Review #1**

**Rating:** 3

**Review:**

This paper introduces an approach to recover weights of ReLU neural networks by querying the network with specifically constructed inputs. The authors notice that the decision regions of such networks are piece-wise linear corresponding to activations of individual neurons. This allows to identify hyperplanes that constitute the decision boundary and find intersection points of the decision boundaries corresponding to neurons at different layers of the network. However, weights can be recovered only up to permutations of neurons in each layer and up to a constant scaling factor for each layer.

The algorithm consists of two parts: Identifying parameters of the first layer and subsequent layers. First they sample a lot of line segments and find their intersections with the decision boundary, i.e. where pre-activations equal 0. Then they sample some points around the intersections and estimate the hyperplanes up to a scaling factor and a sign. For the first layer they check whether some of the hyperplanes belong to it. For the consecutive layers they proceed by moving from the intersection points along the hyperplanes until the decision boundary bends. Again, by identifying which of the bends correspond to the intersection of the current layer's hyperplanes with the previous layers' ones, they are able to recover parameters of the current layer by computing the angles of these intersections.

This paper tackles a very interesting and important problem that might have huge implications for security and many other aspects. However, I'm leaning towards Reject for the following reasons:

1. The algorithm's description is either incomplete or unclear. There are such core functions as PointsOnLine and TestHyperplane, whose pseudo-code would be very helpful for understanding. For example, the authors say that PointsOnLine performs a "binary search," but then this function can find only one (arbitrary) intersection of a line segment with a decision boundary, while each sampled line can intersect multiple ones. If it is not binary search, then the asymptotic analysis given in the end of Sec. 4.2 is incorrect. Even more mysterious is TestHyperplane, from the provided intuition I do not understand how it is possible to distinguish hyperplanes corresponding to the first layer vs. the other layers. In Sec. 4.3, second paragraph, the choice of R is unclear. How to chose it to make sure that the closest boundary intersects it?

The authors consider a very limited setting of only fully-connected (linear) layers with ReLU activations. In this case it is easy to see that the resulting decision boundary is indeed piece-wise linear with a lot of bending. Authors themselves notice, that "the algorithm does not account for weight sharing." For CNN this will lead to each virtual neuron in each channel to have its own kernel weights, although there must be one kernel per channel. Also the authors admit, that pooling layers affect partitioning of the activation regions, making the proposed approach inapplicable. The authors did not discuss whether the proposed approach can handle batchnorm layers. Such non-linear transformations could pose serious problems. All this rules out applications to, for example, all CNN-based architectures, that prevail in computer vision. The authors mention, that their "algorithm holds with slight modification" for ResNets, but as mentioned earlier convolutional, pooling and batchnorm layers make it not so trivial (if at all possible).

2. Experimental evaluation is extremely limited: It is all contained in just one paragraph. Although it is mentioned that "it is often possible to recover parameters of deep ReLU networks," they evaluated their approach on very shallow and narrow networks (only 2 layers, 10 to 50 neurons in each). The immediate question here is why this algorithm is not applied to sufficiently deep NN? At least a network that could classify MNIST reasonably well. Actually, this would be a better proof-of-concept: Given a pre-trained MNIST classifier, apply the proposed method, recover the weights and check if you get the same output as from the original network. Whereas here the evaluation is given as a normalized relative error of the estimated vs. the true weights. Which raises the question of how the scaling factor was chosen? Recall, that the proposed method estimates network's parameters only up to an arbitrary scaling factor. My guess, is that for the Figures 3 and 4 (both right) the estimated weights were re-scaled optimally to minimize the relative error. But in the end, one is interested in recovering the original weights of the network, not relative ones.

I am very confused by Fig. 3 left: Why is the number of queries going down as the number of neurons increases? Should it not be that with more neurons the ambiguity also increases, requiring more queries? Again, this analysis is very limited, it would be very interesting to see, how many more queries one needs for deeper layers of the network. But for this experiments with deeper than 2 layers networks are necessary.

3. The choice of parameters is unclear and not discussed. How long should the line segments be, how many of them. How many points are sampled and within which radius to identify hyperplanes, how to choose 'R'. And how all these choices affect accuracy and performance.

Overall, the paper looks rather incomplete to me and requires a major revision. It will definitely benefit if the "slight modification" for the case of ResNets is included. Also, experimental evaluation should be completely re-done and extended.

**Experience Assessment:**

I have read many papers in this area.

**Review Assessment: Checking Correctness Of Derivations And Theory:**

I assessed the sensibility of the derivations and theory.

**Review Assessment: Checking Correctness Of Experiments:**

I carefully checked the experiments.

**Review Assessment: Thoroughness In Paper Reading:**

I read the paper at least twice and used my best judgement in assessing the paper.

---

> ### Author Response · Authors · 2019-11-14
> **Response to Review #1**
>
> Thank you for the careful review and feedback.  To respond to the questions raised:
>
> - Detail on algorithmic primitives. We have clarified the text. The algorithm PointsOnLine is able to perform binary search on multiple points simultaneously. Our asymptotic analysis is correct because we know the expected number of boundary points that will be discovered along the line is linear in the total number of neurons of the network (Hanin and Rolnick 2019). For TestHyperplane, the goal is to determine whether all points on a hyperplane do lie within the boundary (since then the hyperplane arises from the boundary of a layer-1 neuron); to do this, we test random points far along the hyperplane - if all these points are indeed on the boundary, then we conclude that the hyperplane is contained in the boundary.
>
> - ResNets. We apologize for the lack of clarity, which we have corrected in the text; we intended to emphasize that our algorithm can be modified to learn skip connections, rather than specialty layers that may also occur in a ResNet. Here is the intuition for such modification: In the case of skip connections, each boundary is still given by a bent hyperplane which bends when it intersects the bent hyperplanes associated with neurons at earlier layers. However, potential weights must in this case be considered between any two neurons in different layers. Deriving such skip weights is somewhat more complex than for MLPs, as the “bend” is influenced not merely by the skip connection but by the weights along all other paths between the two neurons through the network. Thus, it is necessary to “move backward” through the network - for a neuron in layer k, one must first derive the weights in the preceding layer k-1, then at k-2, and so on. If the reviewers believe that such intuition would not confuse the main argument, we are happy to include it in an appendix.
>
> - Evaluation on more complex networks.  We have now added experimental verification of our first-layer method on MLPs trained on MNIST and on networks with 3 and 4 layers.  Please see Figure 3.
>
> - Scaling of weights. As described in 3.2, it is mathematically impossible for any algorithm to learn the “true” scaling of the weights in a network, since this scaling can be arbitrarily changed without in any way affecting the underlying function. As for how we compared the approximated weights with the true weights, we rescaled both sets of weights vectors to norm 1 for comparison (again, since the scaling is arbitrary).
>
> - Figure 3, # of queries. The number of queries shown in the figure is *per parameter learned* - therefore, for larger networks, the number of queries goes up, but not by as much as the number of parameters inferred goes up.
>
> - Number of queries for additional layers. Depending on the approach taken to explore intersections between hyperplanes, the number of queries required can grow linearly in the number of parameters inferred, as each weight can be inferred by examining a single intersection between boundaries.
>
> - Choice of parameters. All choices of parameters are presented in our publicly available code. The length of line segments used in sampling does not significantly affect the results; nor does the radius.
>
> No prior work has, to our knowledge, been able to deduce even the first layer of an MLP with 2 hidden layers.  We show empirically that our algorithm is able to deduce the first layer of 2-, 3-, and 4-layer networks, as well as the second layer of 2-layer networks. The mathematical justification for our algorithm holds for any number of layers.  We believe that each of these contributions significantly advances the state-of-the-art.

---

### Official Review · AnonReviewer3 · 2019-10-23
**Official Blind Review #2331**

**Rating:** 1

**Review:**


Main contribution of the paper
- The paper proposes a new method to recover the unknown structure of  the network by utilizing the piecewise linearity of ReLU network.
- Some theoretical explanation of the method is provided.

Note & Questions
- As far as the author understands, the algorithm does not suppose a fully black-box condition. By seeing the section 4.1 and 4.2, it seems possible to access neurons in the intermediate layers.
- Also, the proposed method seems to target only a MLP.


Strong-points
- This field is not that thoroughly investigated, and the author proposes a creative method to infer the hidden statistics of the neuron.

Concerns
- Most of all, the information the experiments conveys is too small to convince the argument of the author. The reviewer could not find the dataset they train (in the Experiment section), and the graph only shows the case of two-layered networks. Moreover, the reviewer couldn't find the explanation of the graph, including their legend (for example, Memorization).
The author suggests that this method can be applied to various networks. Still, the reviewer couldn't find any clue that the method actually worked for various settings: different activations, convolutional networks, and so on. More experimental results supporting the argument of the authors are required.
- Assuming that the network was trained by MNIST and we infer the weight of the networks by the proposed method. Can the recovered network classify the number as well? Then, how the accuracy change?
More quantitative results regarding the asking are required.
- Experimental results for more-than-two layered networks should be provided.
- Oh.et.al (https://arxiv.org/abs/1711.01768) proposed a blackbox reverse-engineering method and provided experimental settings as well. The author should clarify the novelty and the strong-points of the works compared to the mentioned work.

Conclusion
- The author proposes a new method to recover the weight and bias of the network.
- The reviewer could not find much clue supporting the author's argument from the experiment section.

inquiries
- See the Concerns parts.

**Experience Assessment:**

I have read many papers in this area.

**Review Assessment: Checking Correctness Of Derivations And Theory:**

I assessed the sensibility of the derivations and theory.

**Review Assessment: Checking Correctness Of Experiments:**

I carefully checked the experiments.

**Review Assessment: Thoroughness In Paper Reading:**

I read the paper thoroughly.

---

> ### Author Response · Authors · 2019-11-14
> **Response to Review #3**
>
> Thank you for the careful review and feedback.  To respond to the questions raised:
>
> - We do suppose a completely black box condition. We first deduce the first layer (and describe mathematically why our algorithm works). We then deduce the next layer recursively using our deduction of the previous layer (and again describe mathematically why our algorithm works). We invite the reviewer to verify in our publicly available code that we are not violating the black box condition in the smallest particular.
>
> - We have clarified the description of our experimental setup in Section 6; in particular, we have spelled out that the memorization task involves training an MLP for 1000 epochs using Adam optimizer on a dataset consisting of 1000 ten-dimensional vectors with i.i.d. random coordinates drawn from a unit Gaussian, given arbitrary binary labels.  The network must memorize these points, in keeping with the literature on memorization and generalization (e.g. Zhang et al. 2016).
>
> - We now, as requested, show the success of our method for a network trained on MNIST. Please see Figure 3.
>
> - We have likewise, as requested, added experimental verification of our method for 3-layered MLPs and also 4-layered MLPs. Please see Figure 3.
>
> - We have added a reference to the work of Oh et al. Thank you for calling this work to our attention.
>
> No prior work has, to our knowledge, been able to deduce even the first layer of an MLP with 2 hidden layers.  We show empirically that our algorithm is able to deduce the first layer of 2-, 3-, and 4-layer networks, as well as the second layer of 2-layer networks. The mathematical justification for our algorithm holds for any number of layers. We believe that each of these contributions significantly advances the state-of-the-art.

---

### Official Review · AnonReviewer2 · 2019-10-29
**Official Blind Review #2**

**Rating:** 6

**Review:**

This paper introduces a procedure for reconstructing the architecture and weights of deep ReLU network, given only the ability to query the network (observe network outputs for a sequence of inputs).  The algorithm takes advantage of the piecewise linearity of ReLU networks and an analysis by [Hanin and Rolnick, 2019b] of the boundaries between linear regions as bent hyperplanes.  The observation that a boundary bends only for other boundaries corresponding to neurons in earlier network layers leads to a recursive layer-by-layer procedure for recovering network parameters.  Experiments show ability to recover both random networks and networks trained for a memorization task.  The method is currently limited to ReLU networks and does not account for any parameter-sharing structure, such as that found in convolutional networks.

The networks used in experiments appear to be substantially smaller (e.g. input/output dimensions on the order of 10 neurons) than those used in real applications.  Is the proposed approach practical to apply to networks used in actual applications?  How does the number of queries per parameter scale? (page 5 mentions sample complexity for recovering the first layer, but it would be helpful to clarify the situation for subsequent layers).

Page 7 states that the proposed algorithm also holds for ResNets, with slight modifications, but defers details to future work.  If the modifications are indeed slight, it would better to include them here as this is an important special case and would increase the potential impact of the paper.

Overall, while the paper does appear to rely heavily on developments made by [Hanin and Rolnick, 2019b], there is a potentially interesting contribution here.  I would appreciate clarification on concerns over practicality and the extension to ResNets.


**Experience Assessment:**

I have read many papers in this area.

**Review Assessment: Checking Correctness Of Derivations And Theory:**

I assessed the sensibility of the derivations and theory.

**Review Assessment: Checking Correctness Of Experiments:**

I assessed the sensibility of the experiments.

**Review Assessment: Thoroughness In Paper Reading:**

I read the paper at least twice and used my best judgement in assessing the paper.

---

> ### Author Response · Authors · 2019-11-14
> **Response to Review #2**
>
> Thank you for the careful review and feedback.  To respond to the questions raised:
>
> - Larger networks.  We have now added experimental verification of our first-layer method on MLPs trained on MNIST and on networks with 3 and 4 layers.  Please see Figure 3.
>
> - Number of queries for additional layers. Depending on the approach taken to explore intersections between hyperplanes, the number of queries required can grow linearly in the number of parameters inferred, as each weight can be inferred by examining a single intersection between boundaries.
>
> - ResNets. In the case of skip connections, each boundary is still given by a bent hyperplane which bends when it intersects the bent hyperplanes associated with neurons at earlier layers. However, potential weights must in this case be considered between any two neurons in different layers. Deriving such skip weights is somewhat more complex than for MLPs, as the “bend” is influenced not merely by the skip connection but by the weights along all other paths between the two neurons through the network. Thus, it is necessary to “move backward” through the network - for a neuron in layer k, one must first derive the weights in the preceding layer k-1, then at k-2, and so on. If the reviewers believe that such intuition would not confuse the main argument, we are happy to include it in an appendix.
>
> No prior work has, to our knowledge, been able to deduce even the first layer of an MLP with 2 hidden layers. We show empirically that our algorithm is able to deduce the first layer of 2-, 3-, and 4-layer networks, as well as the second layer of 2-layer networks. The mathematical justification for our algorithm holds for any number of layers.  We believe that each of these contributions significantly advances the state-of-the-art.

---

### Official Review · AnonReviewer4 · 2019-11-01
**Official Blind Review #4**

**Rating:** 6

**Review:**

In this paper, the authors showed that in many cases it is possible to reconstruct the architecture, weights, and biases of a deep ReLU network given the ability to query the network. The studied problem is very interesting. I have the following questions about this paper:

1. Can the authors provide detailed explanation of Figure 1? For instance start from input (x_1, x_2), and the weight in layer 1 and layer 2, what is the exact form of the function plotted in the middle panel? Also, how the input space is partitioned? I appreciate the authors provide this simple example, but detailed math will help readers to understand this easily.

2. How about the efficiency of the proposed method? Is it NP-hard? I would like to see some analysis of the computational complexity and also some related experimental results.

3. If the ReLU network can be reconstructed, can the input also be reconstructed based on the output? It would be very interesting to show a few example on reconstructing the input. Also, is that possible to even reconstruct the training data based on the released model?

**Experience Assessment:**

I do not know much about this area.

**Review Assessment: Checking Correctness Of Derivations And Theory:**

I did not assess the derivations or theory.

**Review Assessment: Checking Correctness Of Experiments:**

I assessed the sensibility of the experiments.

**Review Assessment: Thoroughness In Paper Reading:**

I made a quick assessment of this paper.

---

> ### Author Response · Authors · 2019-11-14
> **Response to Review #4**
>
> Thank you for the careful review and feedback.  To respond to the questions raised:
>
> 1. We have clarified the presentation of Figure 1. The middle panel of the figure shows the output N(x,y) of the network as a function of the two inputs given to the network N. The function N is defined by the network shown on the left panel (where the weights were chosen randomly according to the standard initialization procedure described in Experiments); the network itself is the most succinct description of this function. Regarding the partition of input space, each part within the partition corresponds to the set of input points on which a particular subset of the ReLUs in the network is active, and crossing between two parts within the partition means that (at least) one neuron flips from active to inactive ReLU. Once again, the simplest closed form expression of this partition is given by the network itself - it is not a regular tiling or any other kind of partition that lends itself to succinct description. In some sense, this is the power of the neural network, that it is able to fit complicated functions that cannot be described in other ways.
>
> 2. As we note in the text (see Section 5), there are cases where our algorithm fails - if certain bent hyperplanes coincide exactly or if the boundaries associated with two neurons never intersect.  The former case is vanishingly unlikely for real networks - in particular, slightly perturbing the weights makes it possible for the algorithm to succeed once again, and one will never encounter such a brittle setting as the result of a noisy learning rule.  In the latter case, as we describe in the text, it is possible for the “failure” of the algorithm to actually be a consequence of the network being ill-determined from the start, with several possible isomorphic settings of the weights. Regarding computational complexity, we include several related hardness results in our Related Work section (e.g. Goel, Kanade, et al. 2017).
>
> 3. Reconstructing the input based on the output depends on the nature of the function in question. For example, if the output is of lower dimension than the input, then any continuous mapping will be non-injective - i.e. it will be impossible to recover the input from the output. In cases where the function is injective, it is an excellent question to ask, but one which is likely unrelated to the methods we propose here. Regarding reconstructing the training data based on the model, there is extensive interesting literature on membership inference attacks, such as Shokri et al. 2017, Song et al. 2017, and Carlini et al. 2019.
>
> We have now added experimental verification of our first-layer method on MLPs trained on MNIST and on networks with 3 and 4 layers.  Please see Figure 3. No prior work has, to our knowledge, been able to deduce even the first layer of an MLP with 2 hidden layers.  We show empirically that our algorithm is able to deduce the first layer of 2-, 3-, and 4-layer networks, as well as the second layer of 2-layer networks. The mathematical justification for our algorithm holds for any number of layers.  We believe that each of these contributions significantly advances the state-of-the-art.

---

> > ### Comment · AnonReviewer4 · 2019-11-14
> > **Thank you for you reply**
> >
> > I have read your rebuttal, and most of my questions are well addressed. I maintain my original rating on this paper.

---

### Public Comment · ~Nicholas_Carlini1 · 2019-10-15
**Query efficiency of the algorithm**

Thank you very much for making the code of your algorithm available! It really helps to make the algorithm understandable.

When I run the code to identify the weights of a network with an architecture 10-20-30-1 (so that there are 20 units in the first ReLU layer, and 30 units in the second ReLU layer) it takes 70 million queries to identifying the second layer.

Is this to be expected? For context, this is roughly a hundred thousand queries per trainable parameter. In contrast, it takes under 100,000 queries for the first layer (~400 queries per parameter, in line with Figure 3).

(You may also be interested in https://arxiv.org/abs/1807.05185 which gives a very similar algorithm to yours for the case of one-layer neural networks.)

---

> ### Author Response · Authors · 2019-10-16
> **Query efficiency at second hidden layer**
>
> Great question. Our implementation is not intended to optimize for the number of queries - its efficiency can be improved greatly at the expense of simplicity.  For example, in the current implementation, some neurons in the second layer are estimated repeatedly - using several different points on their associated boundaries - before their full weight vectors are determined.  Sharing information between these iterations would reduce the number of queries needed.
>
> A more subtle optimization approach would have an even greater effect: Suppose that boundary B bends when it intersects boundary B', so that B is given by the two hyperplanes H_1 and H_2 on the two sides of B'.  If H_1 and B' are known, then H_2 is actually almost completely known already - since the "bend" occurs along the intersection of H_1 and B'.  Only a single scalar needs to be determined. Our implementation recalculates the entire hyperplane H_2, but this is in fact unnecessary and leads to a great increase in queries if the input dimension is high.  Reusing this information is straightforward and should certainly be included if the goal is speed over simplicity.
>
> We did not know of the paper you mention until after submission, and will in our revision describe their approach for one-layer ReLU networks - as well your excellent recent paper: https://arxiv.org/abs/1909.01838

---

### Decision · Program_Chairs · 2019-12-19

**Decision:**

Reject

**Comment:**

This article studies the identifiability of architecture and weights of a ReLU network from the values of the computed functions, and presents an algorithm to do this. This is a very interesting problem with diverse implications. The reviewers raised concerns about the completeness of various parts of the proposed algorithm and the complexity analysis, some of which were addressed in the author's response. Another concern raised was that the experiments were limited to small networks, with a proof of concept on more realistic networks missing. The revision added experiments with MNIST. Other concerns (which in my opinion could be studied separately) include possible limitations of the approach to networks with no shared weights nor pooling. The reviewers agree that the article concerns an interesting topic that has not been studied in much detail yet. Still, the article would benefit from a more transparent presentation of the algorithm and theoretical analysis, as well as more extensive experiments.